# BIASED BINARY ATTRIBUTE CLASSIFIERS IGNORE THE MAJORITY CLASSES

## ABSTRACT

To visualize the regions of interest that classifiers base their decisions on, different Class Activation Mapping (CAM) methods have been developed. However, all of these techniques target categorical classifiers only, though most real-world tasks are binary classification. In this paper, we extend gradient-based CAM techniques to work with binary classifiers and visualize the active regions for binary attribute classifiers. When training an unbalanced binary classifier on a biased dataset, it is well-known that the majority class is mostly predicted much better than minority class. In our experiments on the CelebA dataset, we verify these results, when training an unbalanced classifier to extract 40 facial attributes simultaneously. One would expect that the biased classifier has learned to extract features mainly for the majority classes and that the proportional energy of the activations mainly reside in certain specific regions of the image where the attribute is located. However, we find very little regular activation for samples of majority classes, while the active regions for minority classes seem mostly reasonable and overlap with our expectations. These results suggest that biased classifiers only rely on bias activation for majority classes. When training a balanced classifier on the unbalanced data by employing attribute-specific class weights, positive and negative classes are classified similarly well and show expected activations for almost all attributes.

## 1 INTRODUCTION

Binary classification tasks are prevalent in many applications. Unfortunately, many binary classification datasets are highly unbalanced, i.e., one of the two classes appears much more often than the other. When training a classifier on such a biased dataset, it has been shown that the classifier mainly learns the majority class and predicts poorly on the minority class (Rudd et al., 2016). Our experiments validate this behavior. Since the classifier sees much more samples of the majority class during training, one would expect that it learns the features required to classify this class very well.

To assess whether such an assumption actually holds, we make use of techniques for interpretability. Particularly, visualization techniques such as the family of Class Activation Mapping (CAM) methods (Zhou et al., 2016) have been used to analyze the input regions of images that contribute most to the classification. Many of these techniques make use of the network gradients (Selvaraju et al., 2017; Chattopadhay et al., 2018; Draelos & Carin, 2020) to improve the predictions. Yet, these CAM techniques are designed for categorical classifiers, *i. e.*, where more than two classes are predicted, most of them focus on activations resulting from SoftMax. For binary classification tasks, however, only one output is available that presents the prediction for the positive class. Therefore, most categorical classifiers can only highlight the activation of the *positive* class, whereas for binary classifiers it is more important to highlight the *predicted* class. To achieve this, a small modification has to be applied to gradient-based CAM techniques, which we will present in this paper.

Since the CelebA dataset (Liu et al., 2015) contains facial attributes with different severity of imbalance, this dataset provides a nice testbed for our experiments. Using our new technique, we visualize facial attributes extracted by the state-of-the-art Alignment-Free Facial Attribute Classifier (AFFACT) (Günther et al., 2017). This classifier is trained on the raw CelebA dataset, *i. e.*, without taking its bias into account. For highly imbalanced attributes, one would expect that the classifier learns to extract the most important features from the majority classes, while minority classes con-

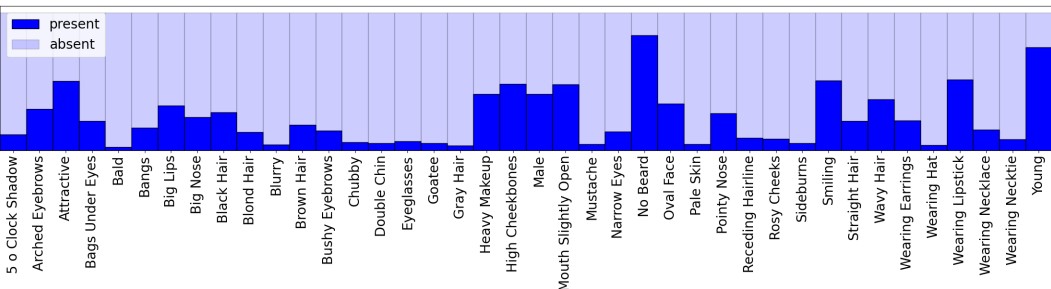

Figure 1: DISTRIBUTION OF ATTRIBUTES. *This figure shows the distribution of the binary facial attributes throughout the CelebA dataset.*

tribute only little to the learned features. Surprisingly, our experiments show the exact opposite: The classification of the majority class is based on the corners of the images or the bias neuron of the final layer. Even worse, this kind of behavior is even propagated to the mostly balanced attributes, since AFFACT learns to predict all attributes simultaneously.

By combining the training method of AFFACT and the Mixed Objective Optimization (MOON) from Rudd et al. (2016), we show that we can train a classifier that is better suited for classifying minority and majority classes, and that this classifier learns to base its predictions on the relevant parts of the images, for both the majority and minority classes.

## 2 RELATED WORK

Many classification tasks throughout the entire field of research are binary, i.e., the model needs to discern between two classes. Examples are identifying man and women (Lin et al., 2016), spam email (Mansoor et al., 2021), malware (Rudd et al., 2017), or skin cancer (Esteva et al., 2017). While some of these tasks are balanced, *i.e.*, both positive and negative classes appear similarly often, many of them are highly biased such that one class appears much more often than the other (Kumar et al., 2022).

### 2.1 FACIAL ATTRIBUTE CLASSIFICATION

One specific task that includes both balanced and unbalanced binary classification tasks is facial attribute prediction. Particularly, the CelebA dataset (Liu et al., 2015) contains 40 binary facial attributes, some of which are balanced (such as `Attractive`) and some are highly unbalanced (such as `Chubby`), as can be seen in Figure 1. Generally, there exist two approaches for facial attribute classification, single-label learning methods which make predictions for each attribute separately, and multi-label learning methods that predict facial attributes concurrently (Mao et al., 2020). While early work (Kumar et al., 2009; Liu et al., 2015; Rozsa et al., 2016; Zhong et al., 2016) relied on single-attribute classifiers, it was realized that combined approaches (Rudd et al., 2016; Hand & Chellappa, 2017; Günther et al., 2017; Zhuang et al., 2018; Rozsa et al., 2019) can leverage from correlations between attributes. While many approaches to jointly classify facial attributes simply ignore the biased nature of some of the attributes (Hand & Chellappa, 2017; Zhuang et al., 2018; Günther et al., 2017), several approaches have been made to provide more balanced attribute classifiers (Rudd et al., 2016; Huang et al., 2016; Kalayeh et al., 2017).

### 2.2 CLASS ACTIVATION MAPPING

To be able to shed some light into the interpretability of machine learning models, there exist several techniques of visualizing the importance of local regions in the input for the final classification (Linardatos et al., 2020). One particular class of methods is based on Class Activation Mapping (CAM) (Zhou et al., 2016), for which several extensions have been proposed (Selvaraju et al., 2017; Chattopadhay et al., 2018; Draelos & Carin, 2020; Wang et al., 2020; Jiang et al., 2021). Many of these methods rely on network gradients and are generally designed to visualize categorical classifiers.

First approaches of visualizing facial attribute classification were performed by Wu et al. (2023), but they visualized a single balanced attribute extracted with a balanced attribute classifier (Rudd et al., 2016) and a single non gradient-based visualization technique (Wang et al., 2020).

## 3 APPROACH

The aim of this paper is to highlight the properties of binary classifiers when trained on unbalanced and balanced datasets. First, we adapt gradient-based CAM techniques to work with binary classifiers. To compare unbalanced and balanced models, we train a balanced model on the unbalanced CelebA dataset. Finally, we evaluate our attribute classifiers by defining regions of the image for each attribute where we would expect the classifier to extract information from, and use these regions to evaluate the interpretability of our classifiers.

### 3.1 VISUALIZING BINARY CLASSIFIERS

Binary classifiers typically use a single output neuron for predicting the presence of the positive class. Oftentimes, a logit:

$$z = \vec{w}^{\mathrm{T}} \vec{\varphi} + b \tag{1}$$

is computed for a given deep feature $\vec{\varphi}$, a learnable fully-connected weight vector $\vec{w}$ and a bias neuron $b$. Afterward, the logit is transformed to a probability using the logistic activation function $y = 1/(1 + e^{-z})$. During inference, the probability $y$ is thresholded at 0.5 to obtain a yes/no answer. Instead, one could also threshold the logit $z$ at 0 to achieve the same result.

Class Activation Mapping (CAM) techniques (Zhou et al., 2016) only work with categorical classifiers, where the contribution for a certain class $c$ shall be predicted. This map estimates the activation at the given spatial location $(i, j)$ by computing a weighted average over the feature map $f_k(i, j)$:

$$A^c(i, j) = \sum_k \alpha_k^c f_k(i, j) \tag{2}$$

where $\alpha_k^c$ is the weight for channel $k$ when classifying class $c$. This activation map $A$ is then rescaled to the input image dimensions. Typically, gradient-based CAM techniques compute these weights by back-propagating the output $y^c$ to the feature map and computing some aggregate of these across locations (Selvaraju et al., 2017):

$$\alpha_k^c = \sum_{(i,j)} \frac{\partial y^c}{\partial f_k(i, j)} \tag{3}$$

The interpretation of this gradient is: In which direction would the feature map need to change in order to increase the probability that class $c$ is predicted?

We apply a similar interpretation for computing the weights for binary classifiers. Here, we take one assumption that includes all binary classifiers trained with logistic activation, but also embraces other loss functions such as the one proposed by Rudd et al. (2016): We threshold the logit score $z$ at 0. Thus, when the classifier predicts the negative class, stronger negative values will increase the prediction of the negative class. Hence, for a binary classifier, we can compute the weight $\alpha_k$ (note that we have only one output here) as:

$$\alpha_k = \sum_{(i,j)} \frac{\partial |z|}{\partial f_k(i, j)} = \sum_{(i,j)} \begin{cases} \frac{\partial z}{\partial f_k(i,j)} & \text{if } z > 0 \\ \frac{\partial(-z)}{\partial f_k(i,j)} & \text{else} \end{cases} \tag{4}$$

This technique can be applied to several gradient-based CAM methods. Compared with traditional Grad-CAM, Grad-CAM++ (Chattopadhay et al., 2018) produces a visual explanation for the class label under consideration by using a weighted mixture of the positive partial derivatives of the last convolutional layer feature maps with respect to a certain class score as weights. Because of its gradient averaging step, Grad-CAM occasionally experiences problems with inaccurate positions. To solve this problem, HiResCAM (Draelos & Carin, 2020), multiplies the activations with the gradients, which can provably guarantee faithfulness for certain models. Element-wise Grad-CAM (Gildenblat & contributors, 2021) is another variant of the Grad-CAM, which multiplies the activations element-wise with the gradients first and then applies a ReLU operation before summing.

In our experiments, we make use of the visualization techniques implemented by Gildenblat & contributors (2021), which allows us to specify the loss function $|z|$ according to (4) for various CAM techniques. We also make use of their default functionality to overlay activations on images. Finally, to show the average activation over several images, we compute the simple average of the overlay images.

## 3.2 Balancing Facial Attribute Classifiers

For training our balanced facial attribute classifier, we combine the two approaches proposed by Rudd et al. (2016) and Günther et al. (2017) to arrive at the balanced AFFACT-b network. To be comparable to the original unbalanced AFFACT network from Günther et al. (2017), which we term AFFACT-u, we use the exact same network, parameters and training schedules. Particularly, we employ a pre-trained ResNet-50 network, which we extend with an additional logit layer to predict 40 facial attributes simultaneously. We apply the same data augmentation as Günther et al. (2017).

For balancing the distributions of attributes, we use the Euclidean loss function (Rudd et al., 2016) averaged over $N$ training samples:

$$\mathcal{J}_w = \frac{1}{N} \sum_{n=1}^{N} \sum_{m=1}^{M} w_m(t_{nm}) \cdot (z_{nm} - t_{nm})^2 \qquad (5)$$

where $m$ represents one of the $M = 40$ different attributes, $t_{nm} \in \{+1, -1\}$ the target label of that attribute, and $z_{nm}$ the prediction of attribute $m$ for sample $n$. For each attribute, we count the probability $p_m$ that a training sample comes from the positive class, and we compute the weight $w_m(t_{nm})$ for the two cases in order to balance the distributions of both classes:

$$p_m = \frac{1}{N} \sum_n \delta_{1,t_{nm}} \quad w_m(+1) = \begin{cases} 1 & \text{if } p_m > 0.5 \\ \frac{1-p_m}{p_m} & \text{else} \end{cases} \quad w_m(-1) = \begin{cases} \frac{p_m}{1-p_m} & \text{if } p_m > 0.5 \\ 1 & \text{else} \end{cases} \qquad (6)$$

where $\delta$ is the Kronecker delta. When assuming a balanced target distribution of classes per attribute, (6) results in the exact same weights as proposed by Rudd et al. (2016). While Rudd et al. (2016) used the weight as a probability to sample whether the loss is applied, we directly apply the weight as a multiplicative factor in (5).

## 3.3 Selecting Frontal Test Images

Since the main focus of our work is the visualization of facial attributes, we only select frontal faces so that a simple aggregation of samples is possible without non-frontal faces disturbing our average CAM results. Also, since most images in CelebA are frontal, we want to exclude random effects arising from non-frontal images that might not have a good representation in the trained models.

For this purpose, we used a simple heuristic on the hand-labeled facial landmarks of the CelebA dataset. Particularly, we computed distance of the nose landmark from the line connecting the center of the mouth corners with the center of the eyes, relative to the distance between eyes center and mouth center. Only faces that had a relative distance smaller than 0.1 were considered as frontal, any other face was excluded from our evaluation. Using this filtering, we obtain 10'458 frontal faces out of the 19'962 CelebA test samples.

## 3.4 Evaluation Metrics

Since we are evaluating balanced and unbalanced binary attribute classifiers, we need to adapt our evaluation technique accordingly. Therefore, we compute False Negative Rates (FNR) and False Positive Rates (FPR) separately per attribute to show which of the two classes is predicted well. Particularly for the unbalanced network, we expect the majority class to be classified well, while the minority class likely has higher error rates.

For evaluating the CAM visualizations, we rely on proportional energy (Wang et al., 2020), which counts how much of the visualization energy is inside a certain binary mask $B$ that we define separately for each attribute. Please refer to the supplemental material for more details on how we

selected these masks. The proportional energy is defined as:

$$E = \frac{\sum\limits_{(i,j)} B(i,j) \cdot A^*(i,j)}{\sum\limits_{(i,j)} A^*(i,j)} \tag{7}$$

where $A^*$ represents the activations $A$ from (2) to input image resolution. We set $E = 0$ when there is no activation $A^*(i,j)$ at any location and, therewith, the denominator vanishes.

## 4 EXPERIMENTS

### 4.1 CLASSIFICATION ERRORS

To verify the expected behavior of our attribute classifiers, we first compute the classification results for the different attributes on our selected frontal images from the test set. The classification results, separated into False Negative Rate, *i. e.*, the number of positively labeled samples wrongly predicted as negative, and False Positive Rate, can be found in Table 1. For easier access, we ordered the attributes by imbalance, starting with the almost balanced attributes, and ending with highly unbalanced ones. The unbalanced original network AFFACT-u works well on both classes as long as the attributes are rather balanced, but already at a small negative/positive imbalance of 68%/32% in the `Wavy Hair` attribute, the prediction of the majority class is about one magnitude better than that of the minority class. For slightly less balanced attributes with about 75%/25% distribution, `Oval Face`, `Pointy Nose` and `Big Lips`, the minority class is even below random performance while the majority class enjoys reasonable classification accuracy.

When using our balancing technique to arrive at the AFFACT-b network, we can observe that False Negative Rates and False Positive Rates are distributed more evenly. Hence, AFFACT-b is able to classify minority and majority classes similarly well, and no error rate goes beyond random chance. However, we observe smaller FNRs than FPRs for many attributes, which might indicate that the presence of a facial attribute is easier to classify than its absence, or that the presence of the attribute is more consistently labeled than its absence, cf. Wu et al. (2023) for a more detailed label analysis.

### 4.2 VISUALIZATION

Having observed that AFFACT-u classifies majority classes well, we expect that this decision is based on reasonable features from the images, while minority class samples with much worse classification performance rely on more dubious features. In Figure 2 we can observe the average activation of our inputs via Grad-CAM, where each pair of images includes the average of all negative predicted attributes on the left, and the average of positive predictions on the right. Again, the attributes are ordered by increasing imbalance, and the visualizations of all attributes can be found in the supplemental. For the most balanced attributes in the first four results from the top row of Figure 2, we can see that AFFACT-u generally makes use of reasonable features, for both classes, and the classification of the presence of an attribute has a larger activated region in the image.

Starting already in the second row and continuing to the bottom, the visualization of the *majority* class tends to rely on the bottom-left corner (sometimes also the other corners) of the image, like `Rosy Cheeks` in Figure 2, or do not show any activation whatsoever like `Bushy Eyebrows` and `Blurry`. The latter can be explained by the fact that the prediction of the majority class solely relies on the bias neuron $b$ in (1) and is not influenced by any feature extracted from the image. When a corner shows activation, we interpret that the network has learned that no relevant features can be extracted from the corners, so these are activated independently of the image input, and they serve as another bias unit similar to $b$. Now, since the network has learned to use the corners as bias units, also more balanced attributes, such as `Attractive`, `Wearing Lipstick` or `High Cheekbones` can assign some energy to these locations that would otherwise be assigned to the bias neuron $b$.

When looking into the visualizations for the AFFACT-b network in Figure 2, one can observe that valid features are extracted for both classes in each attribute, although the minority prediction generally has larger activated regions. This effect can be explained by assuming that minority classes extract stronger features since these samples are weighted higher during training, but it can also be

| Attribute | $p_m$ | Error Rates ↓ | | | | Proportional Energy ↑ | | | |
|---|---|---|---|---|---|---|---|---|---|
| | | AFFACT-u | | AFFACT-b | | AFFACT-u | | AFFACT-b | |
| | | FNR | FPR | FNR | FPR | Pos | Neg | Pos | Neg |
| Attractive | 0.514 | 0.175 | **0.174** | 0.169 | 0.177 | **0.951** | 0.591 | 0.892 | 0.708 |
| Mouth Slightly Open | 0.482 | **0.055** | 0.048 | 0.053 | 0.048 | 0.563 | **0.637** | 0.565 | 0.641 |
| Smiling | 0.480 | **0.072** | 0.063 | 0.065 | 0.064 | 0.556 | **0.547** | 0.554 | 0.651 |
| Wearing Lipstick | 0.470 | **0.065** | 0.037 | 0.054 | 0.043 | 0.406 | **0.374** | 0.458 | 0.446 |
| High Cheekbones | 0.452 | **0.126** | 0.102 | 0.132 | 0.104 | 0.713 | **0.603** | 0.672 | 0.736 |
| Male | 0.419 | **0.017** | 0.008 | 0.017 | 0.008 | 0.986 | **0.889** | 0.997 | 0.981 |
| Heavy Makeup | 0.384 | **0.113** | 0.046 | 0.065 | 0.081 | 0.619 | **0.242** | 0.683 | 0.590 |
| Wavy Hair | 0.319 | **0.258** | 0.058 | 0.187 | 0.093 | 0.470 | **0.096** | 0.472 | 0.201 |
| Oval Face | 0.283 | **0.598** | 0.077 | 0.286 | 0.319 | 0.733 | **0.168** | 0.528 | 0.484 |
| Pointy Nose | 0.276 | **0.629** | 0.055 | 0.336 | 0.209 | 0.543 | **0.253** | 0.520 | 0.447 |
| Arched Eyebrows | 0.266 | **0.274** | 0.110 | 0.119 | 0.221 | 0.621 | **0.064** | 0.686 | 0.309 |
| Big Lips | 0.241 | **0.651** | 0.076 | 0.353 | 0.269 | 0.298 | **0.079** | 0.259 | 0.176 |
| Black Hair | 0.239 | **0.202** | 0.051 | 0.086 | 0.139 | 0.408 | **0.092** | 0.390 | 0.326 |
| Big Nose | 0.236 | **0.367** | 0.115 | 0.165 | 0.266 | 0.621 | **0.113** | 0.596 | 0.417 |
| Young | 0.779 | 0.055 | **0.300** | 0.137 | 0.151 | **0.330** | 0.962 | 0.734 | 0.900 |
| Straight Hair | 0.209 | **0.425** | 0.074 | 0.134 | 0.232 | 0.381 | **0.311** | 0.397 | 0.439 |
| Bags Under Eyes | 0.204 | **0.362** | 0.100 | 0.141 | 0.221 | 0.485 | **0.093** | 0.416 | 0.488 |
| Brown Hair | 0.204 | **0.248** | 0.074 | 0.115 | 0.205 | 0.472 | **0.112** | 0.441 | 0.339 |
| Wearing Earrings | 0.187 | **0.181** | 0.061 | 0.073 | 0.138 | 0.701 | **0.126** | 0.686 | 0.403 |
| No Beard | 0.834 | 0.024 | **0.081** | 0.044 | 0.036 | **0.181** | 0.527 | 0.732 | 0.527 |
| Bangs | 0.152 | **0.119** | 0.020 | 0.038 | 0.053 | 0.764 | **0.013** | 0.704 | 0.729 |
| Blond Hair | 0.149 | **0.153** | 0.017 | 0.048 | 0.067 | 0.332 | **0.018** | 0.298 | 0.223 |
| Bushy Eyebrows | 0.144 | **0.326** | 0.031 | 0.147 | 0.110 | 0.691 | **0.005** | 0.641 | 0.330 |
| Wearing Necklace | 0.121 | **0.507** | 0.038 | 0.152 | 0.230 | 0.633 | **0.020** | 0.580 | 0.302 |
| Narrow Eyes | 0.116 | **0.679** | 0.021 | 0.252 | 0.172 | 0.522 | **0.023** | 0.514 | 0.666 |
| 5 o'Clock Shadow | 0.112 | **0.216** | 0.034 | 0.050 | 0.110 | 0.533 | **0.033** | 0.482 | 0.572 |
| Receding Hairline | 0.080 | **0.398** | 0.026 | 0.092 | 0.129 | 0.716 | **0.004** | 0.724 | 0.528 |
| Wearing Necktie | 0.073 | **0.159** | 0.014 | 0.036 | 0.060 | 0.805 | **0.002** | 0.791 | 0.215 |
| Rosy Cheeks | 0.065 | **0.361** | 0.026 | 0.044 | 0.144 | 0.563 | **0.002** | 0.571 | 0.405 |
| Eyeglasses | 0.065 | **0.018** | 0.002 | 0.013 | 0.007 | 0.710 | **0.001** | 0.770 | 0.571 |
| Goatee | 0.064 | **0.170** | 0.018 | 0.007 | 0.069 | 0.374 | **0.004** | 0.413 | 0.420 |
| Chubby | 0.058 | **0.420** | 0.024 | 0.073 | 0.154 | 0.981 | **0.030** | 0.964 | 0.679 |
| Sideburns | 0.056 | **0.116** | 0.019 | 0.014 | 0.070 | 0.539 | **0.005** | 0.472 | 0.285 |
| Blurry | 0.051 | **0.476** | 0.010 | 0.061 | 0.110 | 0.914 | **0.041** | 0.843 | 0.726 |
| Wearing Hat | 0.049 | **0.078** | 0.004 | 0.028 | 0.015 | 0.886 | **0.002** | 0.872 | 0.622 |
| Double Chin | 0.047 | **0.477** | 0.014 | 0.060 | 0.142 | 0.187 | **0.001** | 0.254 | 0.045 |
| Pale Skin | 0.043 | **0.491** | 0.007 | 0.048 | 0.147 | 0.801 | **0.003** | 0.742 | 0.726 |
| Gray Hair | 0.042 | **0.217** | 0.009 | 0.025 | 0.061 | 0.290 | **0.009** | 0.297 | 0.103 |
| Mustache | 0.041 | **0.505** | 0.011 | 0.034 | 0.081 | 0.421 | **0.004** | 0.512 | 0.573 |
| Bald | 0.023 | **0.191** | 0.005 | 0.025 | 0.032 | 0.744 | **0.001** | 0.687 | 0.431 |

Table 1: ERROR RATES AND PROPORTIONAL ENERGY. *This table shows the probability of the positive class $p_m$, the False Negative and False Positive Rates, as well as Proportional Energy of Grad-CAM visualizations for positively and negatively predicted samples for the different attributes when extracted with an unbalanced and a balanced classifier. The attributes are sorted by increasing imbalance (deviation of $p_m$ from 0.5). For AFFACT-u, the classification error of the minority class, and the proportional energy of the majority class are bolded.*

an effect of the dataset where the minority classes represent the presence of attributes, and the presence generally can rely on a larger set of features then predicting absence of attributes. Anyway, in no case there is any activation in the corners of the images, so the network has successfully learned to ignore the corners that do not include useful information for the classification of any attribute.

## 4.3 PROPORTIONAL ENERGY

To provide a numerical evaluation of the visualizations, we make use of the proportional energy (7) that we compute using the masks defined in the supplemental material. Since the size of the masks differs between attributes, the absolute values of proportional energy cannot be compared across

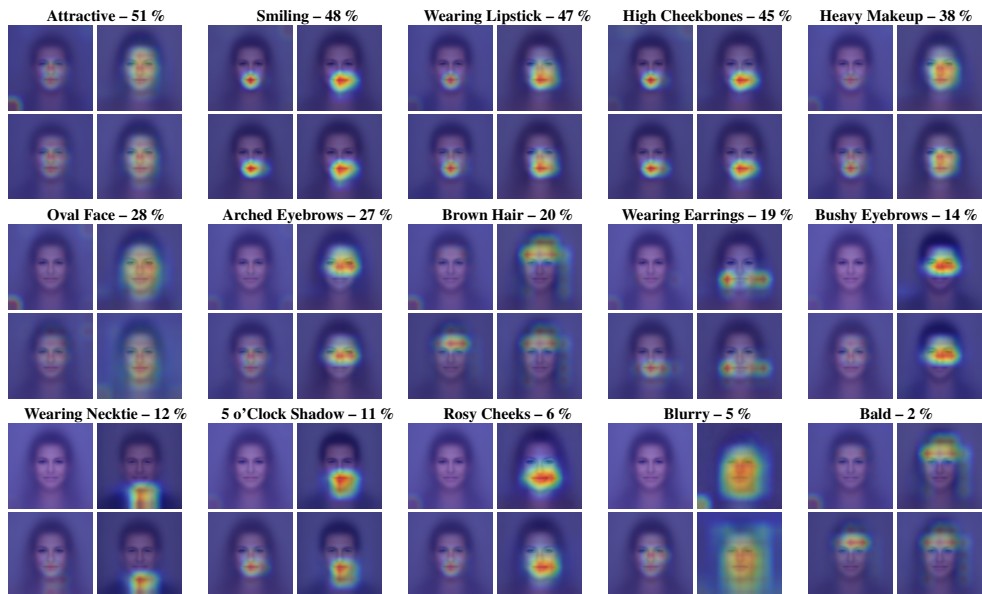

Figure 2: AVERAGED GRAD-CAM ACTIVATIONS. *This figure displays the average CAM activations for 15 different attributes including the probability of positive label $p_m$. Activations are averaged across all negative (left) and positive (right) predictions, extracted by AFFACT-u (top) and AFFACT-b (bottom).*

| Attribute | Method | AFFACT-u | | AFFACT-b | |
|---|---|---|---|---|---|
| | | Pos | Neg | Pos | Neg |
| Attractive | GradCAM | 0.951 | 0.591 | 0.892 | 0.728 |
| | Grad-CAM++ | 0.063 | 0.214 | 0.049 | 0.121 |
| | HiResCAM | 0.977 | 0.784 | 0.966 | 0.916 |
| | Element-wise Grad-CAM | 0.954 | **0.889** | 0.948 | **0.940** |
| Wearing Earrings | GradCAM | 0.701 | 0.126 | 0.686 | 0.403 |
| | Grad-CAM++ | 0.840 | 0.075 | 0.578 | 0.116 |
| | HiResCAM | 0.756 | 0.294 | 0.647 | **0.445** |
| | Element-wise Grad-CAM | 0.631 | **0.326** | 0.504 | 0.401 |
| Bald | GradCAM | 0.744 | 0.001 | 0.687 | **0.431** |
| | Grad-CAM++ | 0.824 | 0.000 | 0.835 | 0.108 |
| | HiResCAM | 0.784 | 0.141 | 0.752 | 0.370 |
| | Element-wise Grad-CAM | 0.699 | **0.158** | 0.540 | 0.311 |

Table 2: PROPORTIONAL ENERGY FOR DIFFERENT TECHNIQUES. *This table shows the Proportional Energy for positively and negatively predicted samples for the different attributes when extracted with an unbalanced and a balanced classifier. The highest proportional energy of the majority class are bolded.*

attributes. Again, we split the results into samples predicted as positive, and samples predicted as negative, and compare the unbalanced and the balanced network. The average proportional energy overall respective samples and for all attributes can be found in Table 1. With these results, we can numerically verify the trend that we could also observe in Figure 2. As soon as the imbalance crosses the 40%/60% border, *i. e.*, starting from `Heavy Makeup` the proportional energy for predicting the majority class via AFFACT-u reduces dramatically when compared to predicting the minority class, which proves strongly that AFFACT-u needs to depend on more dubious features to predict majority classes. For AFFACT-b, there also exist differences in the prediction of presence or absence of features, but these are rarely as pronounced as for AFFACT-u.

Most of these differences highlight that predicting the presence of an attribute might be more localized than predicting its absence, which might include other parts of the face as well. For example, the prediction of the presence of `Wearing Earrings` or `Wearing Necklace` need to focus

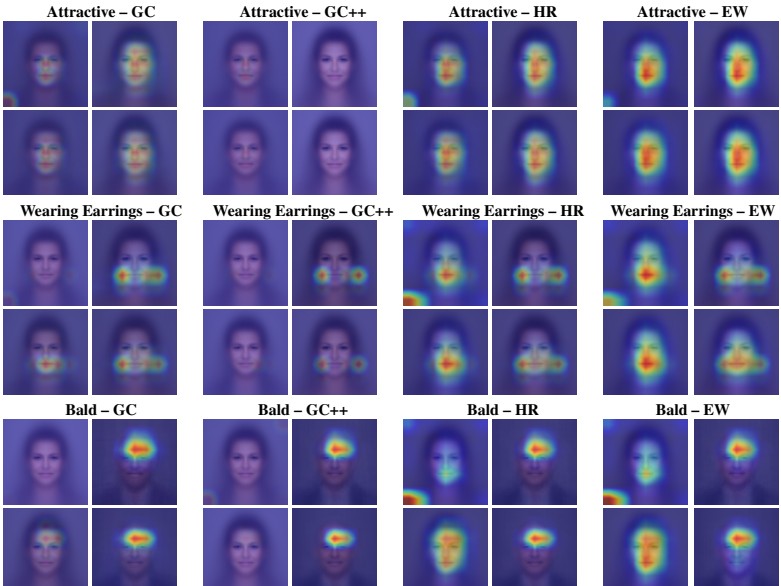

Figure 3: AVERAGED ACTIVATIONS FOR DIFFERENT CAM TECHNIQUES. *This figure shows the average activations for CAM techniques Grad-CAM (GC), GradCAM++ (GC++), HiResCAM (HR) and element-wise CAM (EW). Blocks are built identically to Figure 2.*

more closely to the ear or neck region, while the absence of these can also include locations that indicate the gender of the person – since the presence of such attributes correlate with gender, which can better be approximated from the full face, see the highlighted locations in Figure 2.

## 4.4 CAM TECHNIQUES

When comparing the visualizations of the same attribute from different CAM techniques, we observe that HiResCAM and Element-wise Grad-CAM can visualize the feature maps better for models with negative labels in Figure 3 and the supplemental, especially for AFFACT-u. At the same time, we realize that the results of proportional energy for HiResCAM and Element-wise Grad-CAM are obviously larger than others in Table 2. It indicates that these two techniques can produce visualization results that are clearer and more comprehensive. While HiResCAM and Element-wise CAM include some more information in the prediction of majority classes in AFFACT-u, the corners of the images are still often activated. Additionally, the mouth region is activated for majority classes, independent of the attribute (best seen in the supplemental). GradCAM++ does not extract information of majority classes even for AFFACT-b, we believe that this is an artifact of the visualization technique.

## 4.5 TARGET CLASSES

One of the contributions of this paper is the extension of the categorical classifier visualization to binary classifiers that only have one output node. Here, we show the impact of the visualization when targeting the positive class only, which is what is done in categorical classifiers. Particularly, we use the weight $\alpha_k$ in (4) without computing the absolute value:

$$\alpha_k = \sum_{(i,j)} \frac{\partial z}{\partial f_k(i,j)} \tag{8}$$

Notably, for positively predicted classes, *i.e.* where $z > 0$, both (4) and (8) result in the same visualization. Hence, in Figure 4 we show the impact of our proposed method on negatively-predicted samples for different attributes with various imbalance. To avoid influences of unbalanced predictions discussed in Section 4.2, we utilize our balanced network AFFACT-b in these visualizations. As can be clearly seen in Figure 4, the positive class visualization on the left side of each pair highlights various different regions but the ones that would be expected. Only the predicted class visualization on the right concentrates on the correct part of the image.

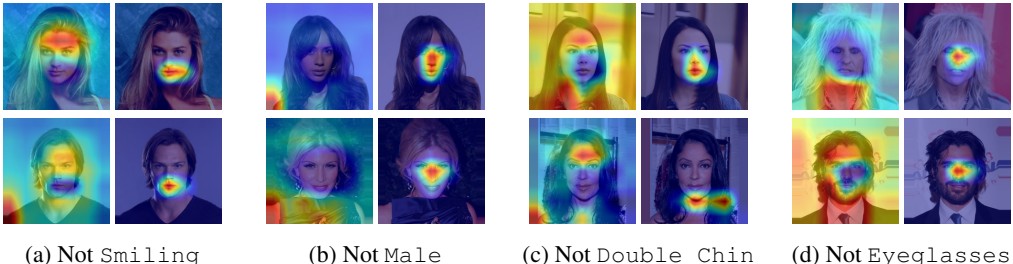

(a) Not `Smiling`    (b) Not `Male`    (c) Not `Double Chin`    (d) Not `Eyeglasses`

Figure 4: AFFACT-B NEGATIVE CLASS VISUALIZATION. *This figure shows Grad-CAM visualizations of samples for four different attributes that were negatively predicted by the balanced network. On the left of each block, we visualize the categorical target via* (8)*, i. e., the positive class. On the right we present the visualization with of the predicted negative class as created with* (4)*.*

## 5  CONCLUSION

In our work, we have modified gradient-based class activation mapping techniques to work with binary classifiers and then applied them to facial attribute classification. We investigated an unbalanced classifier AFFACT-u and showed that this classifier produces extremely low classification errors on majority classes. The visualization results from different CAM techniques prove that these decisions are almost solely based on the bias neuron of the final classification layer or some corners of the image, but not on reasonable areas of the images. On the other hand, minority classes are predicted with extremely high error, sometimes beyond random guessing, but the visualization highlights reasonable regions in the image. Due to the nature of training the classification of several attributes jointly, negative effects from highly unbalanced attributes, i.e., classifying from the corners of the image, are transferred to more balanced attributes.

When applying a training scheme that balances the imbalanced classes, we arrived at the AFFACT-b model, which showed much more reasonable behavior, both in the classification of minority classes (on the cost of misclassifying majority class samples more often), and in the visualization of input regions. While the visualizations in Figure 2 and Figure 3 show only averages, we still observed a few cases where even the balanced classifier has no active regions in the image, especially in highly unbalanced classes, so further research has to be done to understand this corner-case behavior.

### 5.1  DISCUSSION

In this work, we have only used two binary attribute prediction networks, and further studies would be required to validate our findings on other binary classification tasks. Additionally, we have just used one network topology and the Euclidean loss function, but there is no reason to believe that our findings would not translate to other network topologies and binary cross-entropy loss. Also, we have used class weights to provide a balanced network, the influence of FocalLoss (Lin et al., 2017) or other approaches for balancing classes would need further investigation. Besides, we have applied a variety of CAM approaches like GradCAM, Grad-CAM++, HiResCAM, and Element-wise Grad-CAM in our experiments, which provided slightly different views on our conclusion. We also planned to show results for the FullGrad method (Srinivas & Fleuret, 2019), but the available implementation of Gildenblat & contributors (2021) was too slow to run on the large-scale dataset in reasonable time. Finally, our implementation of the binary classifier target only applies to gradient-based CAM techniques, the extension to non gradient-based techniques such as ScoreCAM (Wang et al., 2020) remains unsolved for now. Also, the visualization of some gradient-based methods such as XGradCAM (Fu et al., 2020) do not work with our extension, which needs to be investigated.

For the computation of proportional energy, we have defined some masks that contain reasonable regions in the images. While we have taken care that the masks cover all parts of the image that we deem useful, a better definition of masks will improve proportional energy values for some attributes. However, the overall conclusion in our paper will likely not be influenced by better masks.

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
