# Biased Binary Attribute Classifiers Ignore the Majority Classes

—

# Supplemental Material

## 1 Supplemental

### 1.1 Attribute Masks

Attribute masks were generated based on the fact that most CAM methods work on the final convolutional layer. Since we utilize ResNet-50 as basis, the final convolutional layer reduces the original input image size of $224 \times 224$ pixels to a feature map of resolution $7 \times 7$, each cell of the feature map represents $32 \times 32$ pixels of the input. Thus, we define our masks in terms of $32 \times 32$ blocks, based on the intuition onto which parts of the face the classifier should base its decision on. You can find the masks for the different attributes in Figure 1, some attributes looking into similar regions share the masks. Slightly different masks would likely improve the proportional energy calculation for various attributes, but are out of scope of this work. Notably, none of our masks ever include any corner of the image.

### 1.2 Proportional Energy and Visualization

In the main paper, we had only listed results of a few attributes. The remaining attributes, which we sort by increasing imbalance, can be found here. In Table 1, we show the proportional energy of AFFACT-u and AFFACT-b for four CAM techniques, averaged for positively and negatively predicted attributes. In Figures 2-5, you can find the average activations for these visualizations techniques.

Table 1: `Proportional Energy.` *This table includes Proportional Energy values obtained through four different CAM techniques, averaged for all positively and negatively predicted samples per attribute by two networks.*

| Attribute | $p_m$ | Method | AFFACT-u | | AFFACT-b | |
|---|---|---|---|---|---|---|
| | | | Pos | Neg | Pos | Neg |
| Attractive | 0.514 | GradCAM | 0.951 | 0.591 | 0.892 | 0.708 |
| | | Grad-CAM++ | 0.063 | 0.214 | 0.049 | 0.121 |
| | | HiResCAM | 0.977 | 0.784 | 0.966 | 0.916 |
| | | Element-wise CAM | 0.954 | 0.889 | 0.948 | 0.940 |
| Mouth Sl. Open | 0.482 | GradCAM | 0.563 | 0.637 | 0.565 | 0.641 |
| | | Grad-CAM++ | 0.647 | 0.715 | 0.648 | 0.705 |
| | | HiResCAM | 0.597 | 0.412 | 0.563 | 0.489 |
| | | Element-wise CAM | 0.501 | 0.393 | 0.456 | 0.409 |
| Smiling | 0.480 | GradCAM | 0.556 | 0.547 | 0.554 | 0.651 |
| | | Grad-CAM++ | 0.702 | 0.669 | 0.685 | 0.695 |
| | | HiResCAM | 0.614 | 0.342 | 0.540 | 0.481 |
| | | Element-wise CAM | 0.500 | 0.354 | 0.429 | 0.401 |
| Wearing Lipstick | 0.470 | GradCAM | 0.406 | 0.374 | 0.458 | 0.446 |
| | | Grad-CAM++ | 0.447 | 0.265 | 0.404 | 0.141 |
| | | HiResCAM | 0.350 | 0.272 | 0.337 | 0.303 |
| | | Element-wise CAM | 0.324 | 0.276 | 0.294 | 0.273 |

Table 1: (Continued)

| Attribute | $p_m$ | Method | AFFACT-u | | AFFACT-b | |
|---|---|---|---|---|---|---|
| | | | Pos | Neg | Pos | Neg |
| High Cheekbones | 0.452 | GradCAM | 0.713 | 0.603 | 0.672 | 0.736 |
| | | Grad-CAM++ | 0.833 | 0.673 | 0.722 | 0.474 |
| | | HiResCAM | 0.698 | 0.481 | 0.639 | 0.609 |
| | | Element-wise CAM | 0.604 | 0.513 | 0.558 | 0.546 |
| Male | 0.419 | GradCAM | 0.986 | 0.889 | 0.997 | 0.981 |
| | | Grad-CAM++ | 0.863 | 0.553 | 0.742 | 0.846 |
| | | HiResCAM | 0.979 | 0.962 | 0.985 | 0.976 |
| | | Element-wise CAM | 0.964 | 0.944 | 0.973 | 0.963 |
| Heavy Makeup | 0.384 | GradCAM | 0.619 | 0.242 | 0.683 | 0.590 |
| | | Grad-CAM++ | 0.585 | 0.227 | 0.475 | 0.216 |
| | | HiResCAM | 0.632 | 0.419 | 0.599 | 0.562 |
| | | Element-wise CAM | 0.578 | 0.459 | 0.541 | 0.508 |
| Wavy Hair | 0.319 | GradCAM | 0.470 | 0.096 | 0.472 | 0.201 |
| | | Grad-CAM++ | 0.505 | 0.015 | 0.502 | 0.063 |
| | | HiResCAM | 0.474 | 0.187 | 0.417 | 0.248 |
| | | Element-wise CAM | 0.387 | 0.204 | 0.319 | 0.240 |
| Oval Face | 0.283 | GradCAM | 0.733 | 0.168 | 0.528 | 0.484 |
| | | Grad-CAM++ | 0.070 | 0.014 | 0.009 | 0.037 |
| | | HiResCAM | 0.752 | 0.387 | 0.696 | 0.558 |
| | | Element-wise CAM | 0.739 | 0.633 | 0.729 | 0.719 |
| Pointy Nose | 0.276 | GradCAM | 0.543 | 0.253 | 0.520 | 0.447 |
| | | Grad-CAM++ | 0.571 | 0.021 | 0.056 | 0.110 |
| | | HiResCAM | 0.704 | 0.337 | 0.530 | 0.472 |
| | | Element-wise CAM | 0.576 | 0.419 | 0.476 | 0.464 |
| Arched Eyebrows | 0.266 | GradCAM | 0.621 | 0.064 | 0.686 | 0.309 |
| | | Grad-CAM++ | 0.764 | 0.019 | 0.400 | 0.149 |
| | | HiResCAM | 0.702 | 0.242 | 0.599 | 0.403 |
| | | Element-wise CAM | 0.560 | 0.322 | 0.485 | 0.418 |
| Big Lips | 0.241 | GradCAM | 0.298 | 0.079 | 0.259 | 0.176 |
| | | Grad-CAM++ | 0.200 | 0.008 | 0.012 | 0.007 |
| | | HiResCAM | 0.359 | 0.128 | 0.265 | 0.171 |
| | | Element-wise CAM | 0.284 | 0.202 | 0.233 | 0.208 |
| Black Hair | 0.239 | GradCAM | 0.408 | 0.092 | 0.390 | 0.326 |
| | | Grad-CAM++ | 0.381 | 0.011 | 0.253 | 0.142 |
| | | HiResCAM | 0.400 | 0.263 | 0.366 | 0.302 |
| | | Element-wise CAM | 0.341 | 0.250 | 0.291 | 0.267 |
| Big Nose | 0.236 | GradCAM | 0.621 | 0.113 | 0.596 | 0.417 |
| | | Grad-CAM++ | 0.647 | 0.013 | 0.060 | 0.036 |
| | | HiResCAM | 0.674 | 0.265 | 0.574 | 0.414 |
| | | Element-wise CAM | 0.562 | 0.381 | 0.491 | 0.442 |
| Young | 0.779 | GradCAM | 0.330 | 0.962 | 0.734 | 0.900 |
| | | Grad-CAM++ | 0.027 | 0.698 | 0.033 | 0.097 |
| | | HiResCAM | 0.811 | 0.988 | 0.947 | 0.947 |
| | | Element-wise CAM | 0.875 | 0.972 | 0.940 | 0.951 |
| Straight Hair | 0.209 | GradCAM | 0.381 | 0.311 | 0.397 | 0.439 |
| | | Grad-CAM++ | 0.384 | 0.144 | 0.185 | 0.309 |
| | | HiResCAM | 0.378 | 0.266 | 0.357 | 0.343 |
| | | Element-wise CAM | 0.321 | 0.255 | 0.279 | 0.278 |
| Bags Under Eyes | 0.204 | GradCAM | 0.485 | 0.093 | 0.416 | 0.488 |
| | | Grad-CAM++ | 0.722 | 0.021 | 0.094 | 0.131 |
| | | HiResCAM | 0.580 | 0.222 | 0.412 | 0.448 |
| | | Element-wise CAM | 0.486 | 0.306 | 0.412 | 0.414 |

Table 1: (Continued)

| Attribute | $p_m$ | Method | AFFACT-u Pos | AFFACT-u Neg | AFFACT-b Pos | AFFACT-b Neg |
|---|---|---|---|---|---|---|
| Brown Hair | 0.204 | GradCAM | 0.472 | 0.112 | 0.441 | 0.339 |
| | | Grad-CAM++ | 0.470 | 0.025 | 0.255 | 0.156 |
| | | HiResCAM | 0.478 | 0.234 | 0.420 | 0.300 |
| | | Element-wise CAM | 0.401 | 0.233 | 0.316 | 0.267 |
| Wearing Earrings | 0.187 | GradCAM | 0.701 | 0.126 | 0.686 | 0.403 |
| | | Grad-CAM++ | 0.840 | 0.075 | 0.578 | 0.116 |
| | | HiResCAM | 0.756 | 0.294 | 0.647 | 0.445 |
| | | Element-wise CAM | 0.631 | 0.326 | 0.504 | 0.401 |
| No Beard | 0.834 | GradCAM | 0.181 | 0.527 | 0.732 | 0.527 |
| | | Grad-CAM++ | 0.096 | 0.572 | 0.426 | 0.576 |
| | | HiResCAM | 0.364 | 0.530 | 0.575 | 0.509 |
| | | Element-wise CAM | 0.396 | 0.527 | 0.517 | 0.495 |
| Bangs | 0.152 | GradCAM | 0.764 | 0.013 | 0.704 | 0.729 |
| | | Grad-CAM++ | 0.833 | 0.005 | 0.818 | 0.448 |
| | | HiResCAM | 0.808 | 0.287 | 0.747 | 0.586 |
| | | Element-wise CAM | 0.737 | 0.312 | 0.625 | 0.502 |
| Blond Hair | 0.149 | GradCAM | 0.332 | 0.018 | 0.298 | 0.223 |
| | | Grad-CAM++ | 0.265 | 0.001 | 0.230 | 0.063 |
| | | HiResCAM | 0.318 | 0.232 | 0.322 | 0.242 |
| | | Element-wise CAM | 0.302 | 0.224 | 0.287 | 0.236 |
| Bushy Eyebrows | 0.144 | GradCAM | 0.691 | 0.005 | 0.641 | 0.330 |
| | | Grad-CAM++ | 0.822 | 0.002 | 0.771 | 0.090 |
| | | HiResCAM | 0.809 | 0.223 | 0.758 | 0.400 |
| | | Element-wise CAM | 0.698 | 0.296 | 0.575 | 0.436 |
| Wearing Necklace | 0.121 | GradCAM | 0.633 | 0.020 | 0.580 | 0.302 |
| | | Grad-CAM++ | 0.842 | 0.000 | 0.305 | 0.025 |
| | | HiResCAM | 0.744 | 0.125 | 0.525 | 0.249 |
| | | Element-wise CAM | 0.514 | 0.148 | 0.313 | 0.211 |
| Narrow Eyes | 0.116 | GradCAM | 0.522 | 0.023 | 0.514 | 0.666 |
| | | Grad-CAM++ | 0.818 | 0.006 | 0.154 | 0.613 |
| | | HiResCAM | 0.792 | 0.263 | 0.423 | 0.680 |
| | | Element-wise CAM | 0.657 | 0.328 | 0.434 | 0.521 |
| 5 o'Clock Shadow | 0.112 | GradCAM | 0.533 | 0.033 | 0.482 | 0.572 |
| | | Grad-CAM++ | 0.608 | 0.011 | 0.388 | 0.187 |
| | | HiResCAM | 0.538 | 0.285 | 0.532 | 0.517 |
| | | Element-wise CAM | 0.517 | 0.346 | 0.495 | 0.487 |
| Receding Hairline | 0.080 | GradCAM | 0.716 | 0.004 | 0.724 | 0.528 |
| | | Grad-CAM++ | 0.843 | 0.001 | 0.797 | 0.078 |
| | | HiResCAM | 0.810 | 0.157 | 0.727 | 0.335 |
| | | Element-wise CAM | 0.665 | 0.189 | 0.506 | 0.307 |
| Wearing Necktie | 0.073 | GradCAM | 0.805 | 0.002 | 0.791 | 0.215 |
| | | Grad-CAM++ | 0.886 | 0.000 | 0.892 | 0.137 |
| | | HiResCAM | 0.863 | 0.130 | 0.810 | 0.232 |
| | | Element-wise CAM | 0.727 | 0.142 | 0.542 | 0.206 |
| Rosy Cheeks | 0.065 | GradCAM | 0.563 | 0.002 | 0.571 | 0.405 |
| | | Grad-CAM++ | 0.793 | 0.000 | 0.589 | 0.149 |
| | | HiResCAM | 0.707 | 0.230 | 0.663 | 0.465 |
| | | Element-wise CAM | 0.635 | 0.303 | 0.552 | 0.477 |
| Eyeglasses | 0.065 | GradCAM | 0.710 | 0.001 | 0.770 | 0.571 |
| | | Grad-CAM++ | 0.823 | 0.000 | 0.820 | 0.276 |
| | | HiResCAM | 0.816 | 0.201 | 0.768 | 0.475 |
| | | Element-wise CAM | 0.750 | 0.221 | 0.630 | 0.442 |

Continued on the next page

Table 1: (Continued)

| Attribute | $p_m$ | Method | AFFACT-u | | AFFACT-b | |
|---|---|---|---|---|---|---|
| | | | Pos | Neg | Pos | Neg |
| Goatee | 0.064 | GradCAM | 0.374 | 0.004 | 0.413 | 0.420 |
| | | Grad-CAM++ | 0.495 | 0.008 | 0.564 | 0.207 |
| | | HiResCAM | 0.444 | 0.116 | 0.439 | 0.244 |
| | | Element-wise CAM | 0.380 | 0.135 | 0.333 | 0.216 |
| Chubby | 0.058 | GradCAM | 0.981 | 0.030 | 0.964 | 0.679 |
| | | Grad-CAM++ | 0.999 | 0.004 | 0.210 | 0.039 |
| | | HiResCAM | 0.993 | 0.688 | 0.967 | 0.951 |
| | | Element-wise CAM | 0.986 | 0.775 | 0.964 | 0.946 |
| Sideburns | 0.056 | GradCAM | 0.539 | 0.005 | 0.472 | 0.285 |
| | | Grad-CAM++ | 0.498 | 0.007 | 0.353 | 0.083 |
| | | HiResCAM | 0.522 | 0.235 | 0.454 | 0.415 |
| | | Element-wise CAM | 0.490 | 0.264 | 0.416 | 0.394 |
| Blurry | 0.051 | GradCAM | 0.914 | 0.041 | 0.843 | 0.726 |
| | | Grad-CAM++ | 0.984 | 0.002 | 0.936 | 0.032 |
| | | HiResCAM | 0.976 | 0.782 | 0.819 | 0.958 |
| | | Element-wise CAM | 0.914 | 0.756 | 0.911 | 0.944 |
| Wearing Hat | 0.049 | GradCAM | 0.886 | 0.002 | 0.872 | 0.622 |
| | | Grad-CAM++ | 0.897 | 0.003 | 0.881 | 0.186 |
| | | HiResCAM | 0.875 | 0.215 | 0.840 | 0.422 |
| | | Element-wise CAM | 0.796 | 0.223 | 0.634 | 0.375 |
| Double Chin | 0.047 | GradCAM | 0.187 | 0.001 | 0.254 | 0.045 |
| | | Grad-CAM++ | 0.159 | 0.000 | 0.080 | 0.005 |
| | | HiResCAM | 0.200 | 0.064 | 0.259 | 0.108 |
| | | Element-wise CAM | 0.176 | 0.081 | 0.170 | 0.120 |
| Pale Skin | 0.043 | GradCAM | 0.801 | 0.003 | 0.742 | 0.726 |
| | | Grad-CAM++ | 0.915 | 0.000 | 0.744 | 0.067 |
| | | HiResCAM | 0.907 | 0.408 | 0.791 | 0.766 |
| | | Element-wise CAM | 0.892 | 0.502 | 0.802 | 0.775 |
| Gray Hair | 0.042 | GradCAM | 0.290 | 0.009 | 0.297 | 0.103 |
| | | Grad-CAM++ | 0.255 | 0.005 | 0.167 | 0.012 |
| | | HiResCAM | 0.278 | 0.228 | 0.270 | 0.216 |
| | | Element-wise CAM | 0.252 | 0.216 | 0.230 | 0.218 |
| Mustache | 0.041 | GradCAM | 0.421 | 0.004 | 0.512 | 0.573 |
| | | Grad-CAM++ | 0.642 | 0.003 | 0.670 | 0.290 |
| | | HiResCAM | 0.574 | 0.152 | 0.469 | 0.358 |
| | | Element-wise CAM | 0.460 | 0.174 | 0.365 | 0.301 |
| Bald | 0.023 | GradCAM | 0.744 | 0.001 | 0.687 | 0.431 |
| | | Grad-CAM++ | 0.824 | 0.000 | 0.835 | 0.108 |
| | | HiResCAM | 0.784 | 0.141 | 0.752 | 0.370 |
| | | Element-wise CAM | 0.699 | 0.158 | 0.540 | 0.311 |

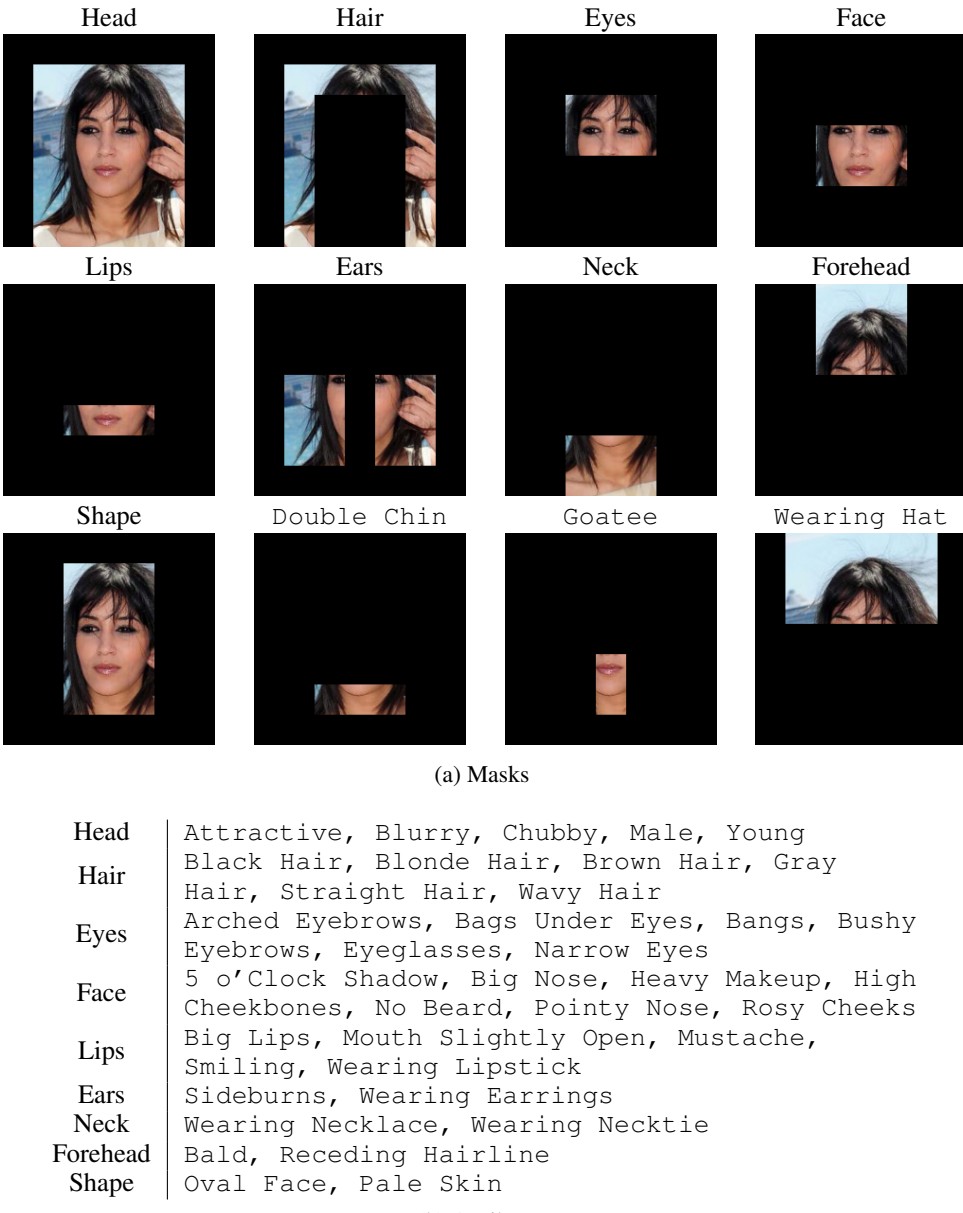

(a) Masks

| Head | Attractive, Blurry, Chubby, Male, Young |
|---|---|
| Hair | Black Hair, Blonde Hair, Brown Hair, Gray Hair, Straight Hair, Wavy Hair |
| Eyes | Arched Eyebrows, Bags Under Eyes, Bangs, Bushy Eyebrows, Eyeglasses, Narrow Eyes |
| Face | 5 o'Clock Shadow, Big Nose, Heavy Makeup, High Cheekbones, No Beard, Pointy Nose, Rosy Cheeks |
| Lips | Big Lips, Mouth Slightly Open, Mustache, Smiling, Wearing Lipstick |
| Ears | Sideburns, Wearing Earrings |
| Neck | Wearing Necklace, Wearing Necktie |
| Forehead | Bald, Receding Hairline |
| Shape | Oval Face, Pale Skin |

(b) Attributes

Figure 1: ATTRIBUTE MASKS. *The images in (a) show the different defined masks, applied to one input image. (b) lists the attributes for which the masks are valid for. The last three masks are defined for single attributes.*

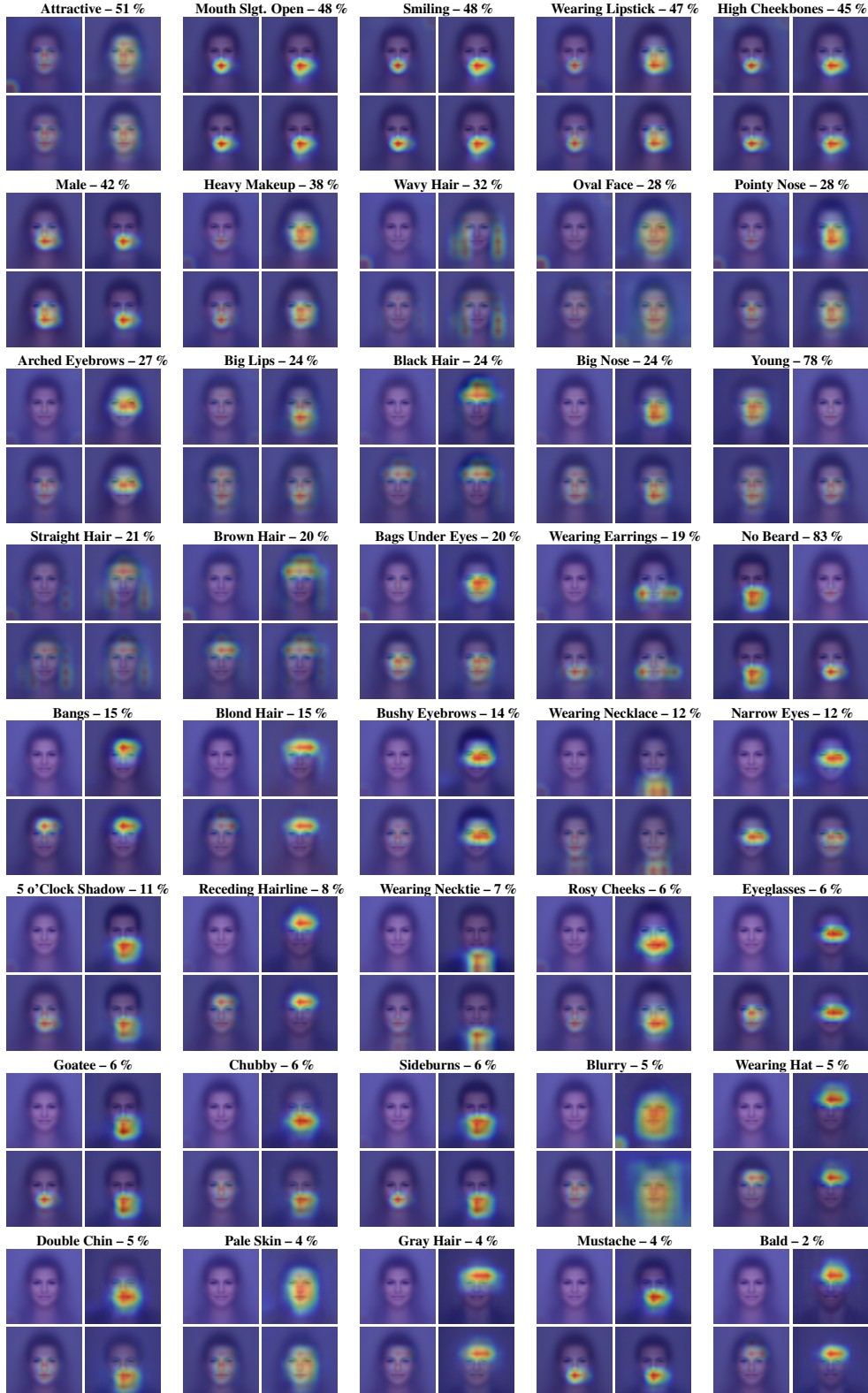

Figure 2: AVERAGED GRAD-CAM ACTIVATIONS. *This figure displays the average CAM activations for 40 different attributes including the probability of positive label $p_m$. Activations are averaged across all negative (left) and positive (right) predictions, extracted by AFFACT-u (top) and AFFACT-b (bottom).*

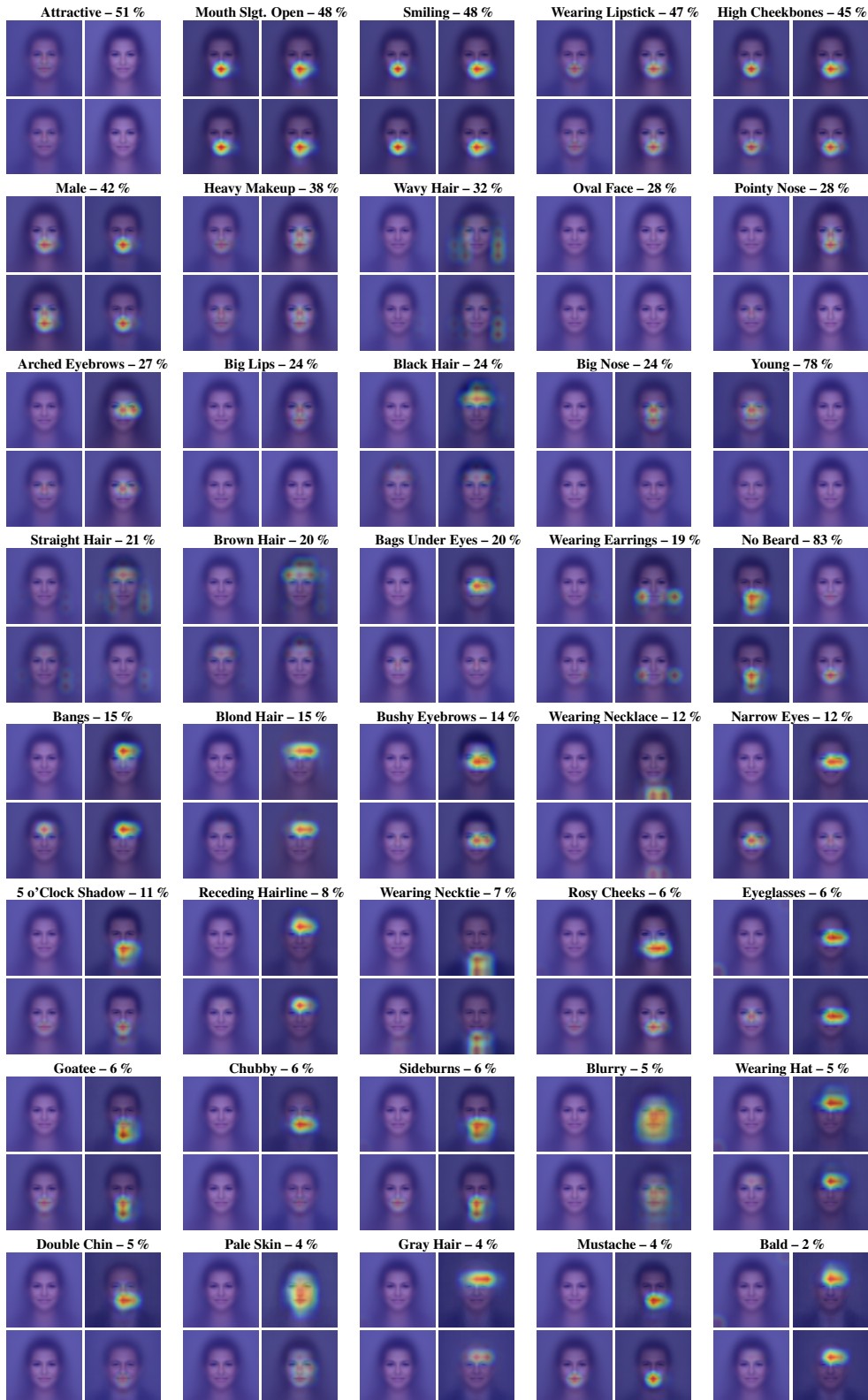

Figure 3: AVERAGED GRADCAM++ ACTIVATIONS. *This figure displays the average CAM activations for 40 different attributes including the probability of positive label $p_m$. Activations are averaged across all negative (left) and positive (right) predictions, extracted by AFFACT-u (top) and AFFACT-b (bottom).*

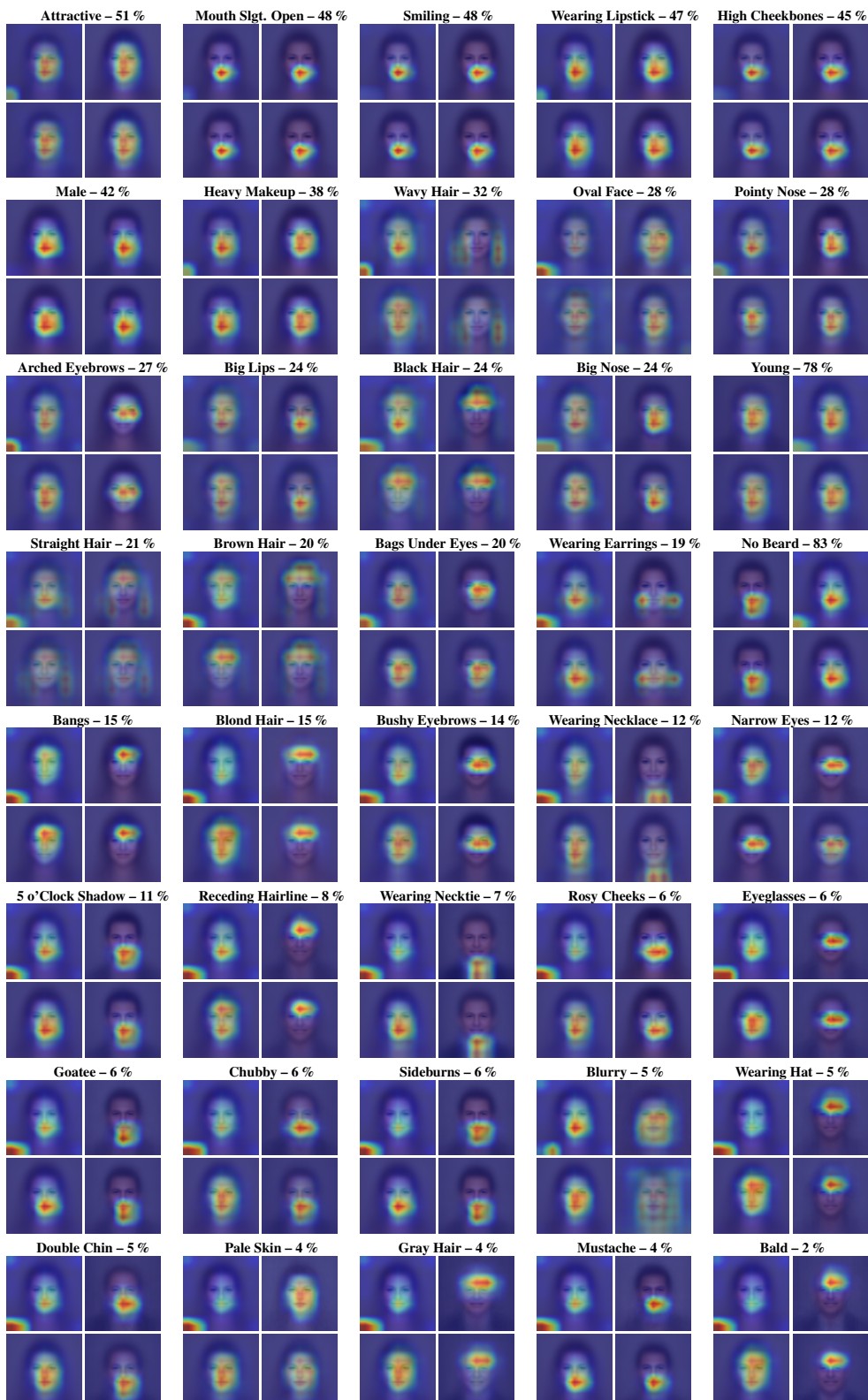

Figure 4: AVERAGED HIRESCAM ACTIVATIONS. *This figure displays the average CAM activations for 40 different attributes including the probability of positive label $p_m$. Activations are averaged across all negative (left) and positive (right) predictions, extracted by AFFACT-u (top) and AFFACT-b (bottom).*

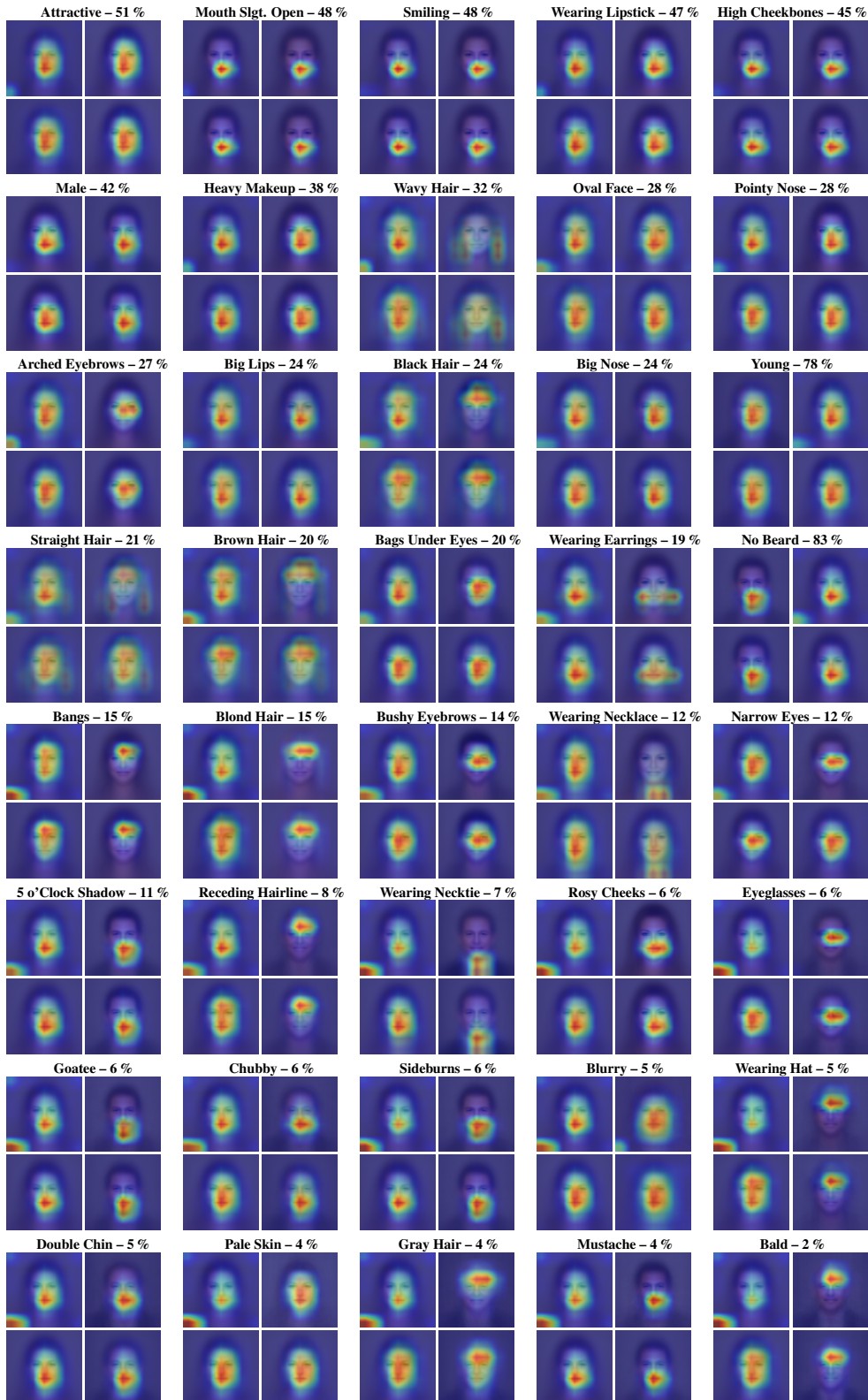

Figure 5: AVERAGED ELEMENT-WISE CAM ACTIVATIONS. *This figure displays the average CAM activations for 40 different attributes including the probability of positive label $p_m$. Activations are averaged across all negative (left) and positive (right) predictions, extracted by AFFACT-u (top) and AFFACT-b (bottom).*