# OpenReview forum: "Biased Binary Attribute Classifiers Ignore the Majority Classes"
_ICLR.cc/2024/Conference — ICLR 2024 Conference Withdrawn Submission_

### Official Review · Reviewer_yjBg · 2023-10-22

**Soundness:** 3 good
**Presentation:** 4 excellent
**Contribution:** 2 fair
**Rating:** 3
**Confidence:** 4

**Summary:**

This paper presents an analysis of the impact of imbalance in the data distribution and implementation of class activation maps for binary classifiers. The authors present their results on the CELEB-A dataset using the AFFACT-u and AFFACT-b classifier. They analyze the proportional energy of the activation corresponding to each label.

**Strengths:**

See summary

**Weaknesses:**

1. Comparisons against methods capable of rebalancing/mitigating the effects of the underlying distribution are lacking. Methods such as logit adjustment Menon et al, Sharpness aware minimization Foret et al, and Rangwani et al to name a few are missing.
2. It would be helpful if the authors could elaborate further on the novelty of their contribution since this appears to not differ significantly from standard grad-cam implementation.
3. How do these aforementioned (1) methods of proportional energy compare against when analyzed with existing methods such as grad cam?


[1] Long-tail learning via logit adjustment by Menon et al
[2] Sharpness-Aware Minimization for Efficiently Improving Generalization Foret et al
[3] Escaping Saddle Points for Effective Generalization on Class-Imbalanced Data by Rangwani et al

**Questions:**

See weakness

---

### Official Review · Reviewer_XzHa · 2023-10-29

**Soundness:** 2 fair
**Presentation:** 3 good
**Contribution:** 2 fair
**Rating:** 3
**Confidence:** 2

**Summary:**

This paper works with the gradient-based class activation mapping (CAM) technique. In particular, the procedure is mainly used with multiclass classifiers and the authors modified it to work with binary classifiers. The authors then investigated an unbalanced classifier AFFACT-u on the Celeb A dataset. First, they verified that the classifier performs well on the majority class for binary classification. However, visualization from CAM techniques show that the reasoning for majority class prediction is not always valid. On the other hand, minority classes are predicted with high error but the reasonable areas of the image get highlighted through CAM. Their proposed classifier AFFACT-b shows better behavior in such a scenario in terms of both classification error and CAM visualization.

**Strengths:**

1. The paper is mostly well written and easy to follow. The authors performed detailed experimentation on the Celeb A dataset to explain their findings.
2. The proposed AFFACT-b classifier clearly performs better than the AFFACT-u classifier on the Celeb A dataset. This can be seen through multiple results, e.g., FNR/FPR, proportional energy and CAM.

**Weaknesses:**

1. The authors bring up some of the weaknesses with respect to the experimentation in the discussion section. For example, they only used two binary attribute prediction networks, they experimented with only one dataset, they have used only one network topology etc. These shortcomings take away from the validity of the current work and further experimental exploration is required.
2. I am not fully convinced by the authors claim "One would expect that the biased classifier has learned to extract features mainly from the majority classes and that the proportional energy of the activations mainly reside in certain specific regions of the image where the attributes is located." In fact, for highly imbalanced binary datasets I would expect the classifier to learn very little about the majority class. Let me take an extreme example where all n/n examples belong to the majority class. In this case, the classifier would see many instances from the "majority" class but wouldn't learn anything about it. Similarly if (n-1) and 1 examples belong to the majority and minority classes respectively, we would expect the classifier to learn very little about the majority class. However, if we put a large weight on the minority class, the classifier might learn about some of its distinguishing features. Thus the finding of biased binary attribute classifiers ignoring the majority classes is not surprising to me.

**Questions:**

1. I would like to know the authors thoughts about the concern raised in Weakness 2.
2. In Section 3.3, the authors mention that they only select frontal test images using a heuristic. I understand the heuristic but some more understanding (maybe examples) would be helpful.

---

### Official Review · Reviewer_uTPu · 2023-10-31

**Soundness:** 2 fair
**Presentation:** 2 fair
**Contribution:** 1 poor
**Rating:** 3
**Confidence:** 4

**Summary:**

In this paper, the authors make slight changes on CAM (Class Activation Mapping) methods to adapt them to the binary classification. Then, they use the resulting technique to visualize the results returned two binary classifiers, AFFACT-u and AFFACT-b classifiers to demonstrate the classification of the majority class is based on the corners of the images or the bias neuron of the final layer.

**Strengths:**

Main strengths of the paper can be summarized as follows:
1) The authors adopt CAM methods for binary classification.
2) The authors introduce AFFACT-b network as an alternative to the classical AFFACT classifier.

**Weaknesses:**

Main weaknesses of the paper can be summarized as follows:
1) The authors argue that CAM techniques are designed for multi-class classifiers and they cannot be used for binary classification using a single logit at the classification layer. Therefore, they introduce a technique to adopt it to such binary classification scenarios. However, for binary classification, one can easily use two class weights that produce two logits without any extra effort (cross entropy loss function using the softmax with two classes), and CAM methods can be used directly. What is the reason for going all this trouble?
2) In general both the novelty and contribution are very limited. Affact-b network uses a well-known loss function from the literature. Also, all findings are built based on AFFACT classifiers which limits the value of the findings. As noted in the discussion, the findings must be also evaluated on different classifiers and loss functions. More precisely, for binary classification, there are one-class type classifiers [R1,R2] using hyperspheres or polyhedral regions that focus on positive data which are more appropriate for imbalanced datasets.
3) I strongly believe that there are problems with the created masks. Such mask must be prepared with qualified psychologists. For example, smiling is not only confined with the lips as given in Supplementary material. Smiling activates regions in the vicinity of cheeks and eyes (see https://www.youtube.com/watch?v=0vlJ-8gXMII&ab_channel=NoldusHumanBehavior).
4) The authors use FNR and FPR rates for measuring accuracy. This is unacceptable. For binary classification especially in imbalanced datasets, map scores obtained from precision recall curves or ROC curves must be used. FNR and FPR largely depend on selected thresholds and they can be misleading. For binary classification, the important thing is the order of scores not the actual scores.
References:
[R1] Cevikalp et al., Deep compact polyhedral conic classifier for open and closed set recognition, Pattern recognition, 2021.
[R2] Ruff. et al., Rethinking assumptions in deep anomaly detection, ICML Workshops, 2021.

**Questions:**

I wonder why the authors did not make any evaluation with the binary classifier using two class weights yielding two logits. In that case, one ca directly use CAM methods.

---

### Official Review · Reviewer_rwvH · 2023-11-01

**Soundness:** 3 good
**Presentation:** 3 good
**Contribution:** 2 fair
**Rating:** 3
**Confidence:** 4

**Summary:**

The paper aims to highlight the properties of binary classifiers when trained on unbalanced and balanced datasets. The scope of bias is limited to “spurious correlations” among attributes in face datasets. The authors adapt gradient-based CAM techniques to work with binary classifiers and compare imbalanced and balanced models. They also study the interpretability of the proposed method and the baselines via CAMs.

The authors find that biased binary attribute classifiers tend to ignore the majority classes, leading to poor performance in these classes, and that balanced models perform better in the majority classes and achieve comparable performance in the minority classes. Their work localises the source of bias to the “bias neuron” of the final output layer.

The authors conclude that it is essential to balance the dataset to improve the performance of binary classifiers. They also suggest that visualising the regions of interest can help improve the interpretability of binary classifiers.

**Strengths:**

1. This work addresses the shortcomings of GradCAM in binary classification tasks.
2. The authors have discovered that attributes from majority classes are not necessarily learnt better, which is an intriguing observation contrary to popular beliefs.
3. The experiments reveal the issues in off-the-self classifiers (AFFACT) with the help of CAMs.
4. The authors have identified potential improvements in [1] by using the loss of [2].


        [1] Günther, Manuel, Andras Rozsa, and Terranee E. Boult. "Affact: Alignment-free facial attribute classification technique." 2017 IEEE International Joint Conference on Biometrics (IJCB). IEEE, 2017.
        [2] Rudd, Ethan M., Manuel Günther, and Terrance E. Boult. "Moon: A mixed objective optimization network for the recognition of facial attributes." Computer Vision–ECCV 2016: 14th European Conference, Amsterdam, The Netherlands, October 11-14, 2016, Proceedings, Part V 14. Springer International Publishing, 2016.

**Weaknesses:**

1. The overall presentation of the paper could be improved
2. The relationship between modifying GradCAM and balancing the model appears disconnected.
3. This work seems derivative, extending [1] with the help of [2] without any clear explanation of observations.
4. Lack of experiments and proper baselines to benchmark against. AFFACT, being effective, is an old approach. Authors need to compare their method with newer ones to make a statement regarding the existence and relevance of this shortcoming.
5. In Table 1, there’s a tradeoff between FPR and FNR in AFFACT-b, which is not clearly explained. Making the higher scores bold for AFFACT-u makes the table open for misinterpretation.

**Questions:**

1. A binary classifier is a special class of categorical classifiers with two classes. It is more generic to use categorical classifiers in tasks with $>= 2$ classes for consistency, and the loss function of a BCE aligns with that of CE with 2 classes. The motivation behind modifying the GradCAM implementation for binary classifiers is not well-presented.
2. The statement from the abstract, “though most real-world tasks are binary classification” is not well-founded. Binary classification might solve all tasks using a one-vs-rest approach. Still, the utility of multi-class classification is an actively researched and important aspect which cannot be ignored.
3. How the authors arrive at the conclusion that the “bias neuron of the last layer is responsible for inducing low performance in minority classes” is unclear.
4. Is avoiding attention at corners the only improvement that’s made possible in this work? The difference in scores is too negligible to understand.
5. The proposed method also attends to the corners (Blurry, Figure 2), and hence, the effect of the “bias neuron” (if such a thing exists) is still not done away with completely with the suggested approach.
6. Figure 1 is a direct copy-paste of Figure 2 in [2]. Can you cite it?
7. Is the proposed method only applicable to the AFFACT family of models? What happens if we have separate binary classifiers for each attribute?
8. In the set of experiments, the bias is examined mostly in one direction- where the "unbiased" means "presence : absence of an attribute is 1 : 1" and biased means "presence : absence of an attribute is P : 1" (0<P<1).  Do the results hold if bias is studied from other side, that is,  "unbiased" means "presence : absence of an attribute is 1 : 1" and biased means "presence : absence of an attribute is 1 : P" (0<P<1)? The authors can carry out experiments to understand this.
9. What is the need for modifying GradCam for binary classification? We can have two neurons and apply softmax on top of it -- that should work for the traditional GradCam.